# Chromatic Bacteria v.2-A Himar1 Transposon-Based Delivery Vector to Extend the Host Range of a Toolbox to Fluorescently Tag Bacteria

Christian Stocks [1,2], Rudolf O. Schlechter [1,2,3,4] and Mitja N. P. Remus-Emsermann [1,2,3,4,*]

1. School of Biological Sciences, Faculty of Science, University of Canterbury, Christchurch 8140, New Zealand; cst120@uclive.ac.nz (C.S.); r.schlechter.jahn@fu-berlin.de (R.O.S.)
2. Bioprotection Aotearoa, Lincoln University, Lincoln 7647, New Zealand
3. Biomolecular Interaction Centre, Faculty of Science, University of Canterbury, Christchurch 8140, New Zealand
4. Department of Biology, Chemistry, and Pharmacy, Institute for Microbiology, Freie Universität, 14195 Berlin, Germany
* Correspondence: m.remus-emsermann@fu-berlin.de; Tel.: +49-30-838-58031

**Abstract:** A recent publication described the construction and utility of a comprehensive "Chromatic Bacteria" toolbox containing a set of genetic tools that allows for fluorescently tagging a variety of Proteobacteria. In an effort to expand the range of bacteria taggable with the Chromatic Bacteria toolbox, a series of Himar1 transposon vectors was constructed to mediate insertion of fluorescent protein and antibiotic resistant genes. The Himar1 transposon was chosen as it is known to function in a wide range of bacterial species. To test the suitability of the new Himar1 Chromatic Bacteria plasmid derivatives, conjugations were attempted on recently isolated non-model organisms. Although we were unsuccessful in delivering the plasmids into Gram-positive bacterial isolates, we successfully modified previously recalcitrant isolates to the first set of the Chromatic Bacteria toolbox, such as *Sphingomonas* sp. Leaf357 and *Acidovorax* sp. Leaf84. This manuscript reports on the currently available plasmids and transposition success in different bacteria.

**Keywords:** genetic modification; fluorescent labelling; mariner; plasmid

## 1. Introduction

Recently, we described the construction of a comprehensive "Chromatic Bacteria" toolbox containing a set of plasmids, denoted the pMRE series, to fluorescently tag a wide range of bacterial isolates [1]. These plasmid tools were well received by the community and, at the time of writing this article, have been ordered more than 400 times from the non-profit plasmid distribution platform addgene.org in the last 3 years alone (https://www.addgene.org/browse/article/28196767/, accessed on 7 December 2021). Fluorescently tagging bacteria using genetic manipulation is a state-of-the-art technology to study and track bacterial behaviour and physiology in situ and in vitro, such as on plant leaf surfaces and other environments [2–5]. By fluorescently tagging bacteria, it is possible to study them at the micrometer resolution. This is in stark contrast with the meta-omic research which has driven microbiology and microbial ecology during the last decade. Fluorescent tagging allows the study of bacteria at single cell resolution, which gives insight into biofilm formation, bacteria–bacteria and bacteria–host interactions [1].

The original Chromatic Bacteria toolbox employs three different vectors for fluorescent tagging, paired with one of eight fluorescent proteins and additionally one of four combinations of antibiotic resistant cassettes. The three different vectors are based on (i) a broad-host plasmid, (ii) a Tn5 transposon delivery plasmid, and (iii) a Tn7 transposon delivery plasmid. We have determined the host-range of the Chromatic Bacteria toolbox by extensive plasmid conjugation experiments. As a result, it was shown that, even though wide, the host range of the toolset is limited to Proteobacteria [1].

To further expand the host range of the Chromatic toolbox, we incorporated a Himar1 transposon-based delivery system. The Himar1 transposable element belongs to the mariner family of transposons. Mariner transposons can be found throughout eukaryotic and prokaryotic organisms [6,7]. The Himar1 transposon gene was discovered in the horn fly, *Haematobia irritans*, and has been mutated to construct a hyperactive version [8]. The gene encodes for a transposase protein that functions through a cut and paste mechanism and cleaves sequences between thymine adenine (AT) dinucleotide sites. Unlike other transposases such as the Tn*5* transposase, the Himar1 transposase requires no cofactors to be provided in trans to initialise or mediate gene transposition by the host organism [8]. These low requirements for transposition make Himar1 ideal for random insertion mutagenesis and gene delivery into organisms that do not provide such cofactors. The simplicity in mechanism and broad target range of Himar1 shares similarities with the well characterised Mu transposition complex which has been shown to work in Gram-negative and Gram-positive species [9,10]. The Himar1 transposase has been shown to function in Gram-negative bacteria, such as *Pseudomonas fluorescens* and *Flavobacterium johnsoniae* [11,12], and Gram-positive bacteria, including *Staphylococcus aureus, Clostridium perfringens, Streptococcus mutans*, and *Mycobacterium smegmatis* [13–16]. In this study we describe the construction and utility of Himar1-based suicide vectors that allow the fluorescent tagging of bacteria.

## 2. Results

### 2.1. Construction of pMRE-Himar Series

We constructed a total of 23 plasmid vectors carrying different combinations of fluorescent proteins ranging from cyan to near-infrared fluorescence and antibiotic resistances including chloramphenicol, chloramphenicol and gentamicin; chloramphenicol and kanamycin; and chloramphenicol, kanamycin, and erythromycin (Figure 1, Table 1). For fluorescent protein emissions and excitation spectra, please refer to Schlechter et al. (2018) [1].

**Table 1.** Plasmids constructed in this work.

| Name | Fluorescent Protein Gene | Selectable Marker(s) |
|------|--------------------------|----------------------|
| pMRE-Himar-131 | mTurquoise2 | Cm$^R$ |
| pMRE-Himar-133 | sYFP2 | Cm$^R$ |
| pMRE-Himar-134 | mOrange2 | Cm$^R$ |
| pMRE-Himar-135 | mScarlet-I | Cm$^R$ |
| pMRE-Himar-136 | mCardinal | Cm$^R$ |
| pMRE-Himar-137 | mClover3 | Cm$^R$ |
| pMRE-Himar-140 | mTagBFP2 | Cm$^R$, Gm$^R$ |
| pMRE-Himar-141 | mTurquoise2 | Cm$^R$, Gm$^R$ |
| pMRE-Himar-142 | sGFP2 | Cm$^R$, Gm$^R$ |
| pMRE-Himar-143 | sYFP2 | Cm$^R$, Gm$^R$ |
| pMRE-Himar-144 | mOrange2 | Cm$^R$, Gm$^R$ |
| pMRE-Himar-145 | mScarlet-I | Cm$^R$, Gm$^R$ |
| pMRE-Himar-146 | mCardinal | Cm$^R$, Gm$^R$ |
| pMRE-Himar-147 | mClover3 | Cm$^R$, Gm$^R$ |
| pMRE-Himar-151 | mTurquoise2 | Cm$^R$, Km$^R$ |
| pMRE-Himar-153 | sYFP2 | Cm$^R$, Km$^R$ |
| pMRE-Himar-155 | mScarlet-I | Cm$^R$, Km$^R$ |
| pMRE-Himar-157 | mClover3 | Cm$^R$, Km$^R$ |
| pMRE-Himar-171 | mTurquoise2 | Cm$^R$, Em$^R$ |
| pMRE-Himar-174 | mOrange2 | Cm$^R$, Em$^R$ |
| pMRE-Himar-175 | mScarlet-I | Cm$^R$, Em$^R$ |
| pMRE-Himar-176 | mCardinal | Cm$^R$, Em$^R$ |
| pMRE-Himar-177 | mClover3 | Cm$^R$, Em$^R$ |
| pMRE-Himar-191 | mTurquoise2 | Cm$^R$, Km$^R$, Em$^R$ |
| pMRE-Himar-192 | sGFP2 | Cm$^R$, Km$^R$, Em$^R$ |
| pMRE-Himar-193 | sYFP2 | Cm$^R$, Km$^R$, Em$^R$ |
| pMRE-Himar-194 | mOrange2 | Cm$^R$, Km$^R$, Em$^R$ |
| pMRE-Himar-195 | mScarlet-I | Cm$^R$, Km$^R$, Em$^R$ |
| pMRE-Himar-196 | mCardinal | Cm$^R$, Km$^R$, Em$^R$ |
| pMRE-Himar-197 | mClover3 | Cm$^R$, Km$^R$, Em$^R$ |

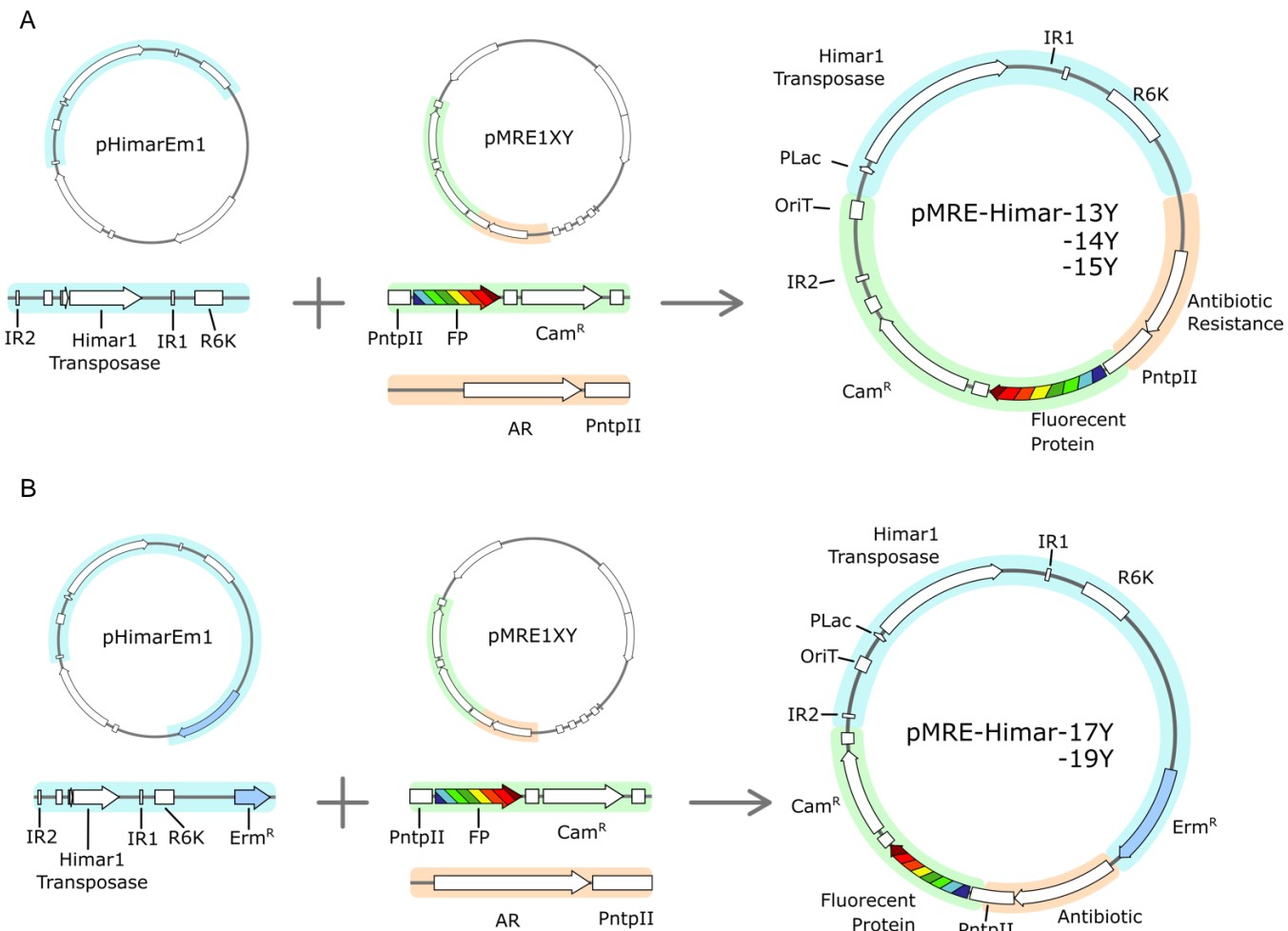

**Figure 1.** Overview of cloning procedures. (**A**) Construction of pMRE-Himar-13Y to pMRE-Himar-15Y. The pHimarEm1 backbone was amplified using PCR and Gibson assembly primers without the erythromycin resistance gene. Plasmids from the original Chromatic Bacteria series were PCR amplified in two separate reactions using Gibson assembly primers targeting the chloramphenicol resistance gene and the promoter driving the respective fluorescent protein gene expression and, where applicable, a secondary antibiotic resistance gene. These three fragments were joined using isothermal assembly. (**B**) Construction of pMRE-Himar-17Y and pMRE-Himar-19Y plasmids. The procedure was similar as described in (**A**), but, in this case, the erythromycin resistance gene encoded on the pHimarEm1 plasmid (highlighted in blue) was included when amplifying the backbone.

## 2.2. Transposon Delivery by Conjugation

Using *E. coli* ST18 as a plasmid donor strain, conjugations were performed into the recipient strains listed in Table 2. It was possible to obtain transposon insertion mutants for four Proteobacterial strains as indicated in Table 2; all mutants exhibited detectable fluorescent protein emission as determined using macroscopical observations at the single-colony level (data not shown) and at the single-cell resolution, as determined using widefield epifluorescence microscopy (Figure 2). The insertion sites for several independent insertion events were determined using arbitrary PCR and sequencing. For *Sphingomonas* sp. Leaf357 and *Acidovorax* sp. Leaf84, most of the transposons were inserted in either intergenic regions or hypothetical protein-coding genes (Table 3). We were not able to obtain insertion mutants for strains belonging to the Actinobacteria or Bacteroidetes phyla listed in Table 2 using conjugation.

**Table 2.** Bacterial strains used in this work.

| Strain | Features, Notes | Growth Medium | Growth Temperature | Transconjugants (Yes/No) | Source |
|---|---|---|---|---|---|
| *Escherichia coli* S17-1 | Cloning host for R6K replicon plasmids; :RP4-2 *pro thi hsdR*+ Tp^r Sm^r Tc::Mu-Kan::Tn7/*λpir* | LB | 37 °C | n.a. | [17] |
| *Escherichia coli* ST18 | Conjugation donor; Genotype: S17-1 *λpirΔhemA* | LB, 5-ala | 37 °C | n.a. | [18] |
| *Aeromicrobium* sp. Leaf245 | Transposon recipient (Actinobacteria) | NB | 30 °C | no | [19] |
| *Agreia* sp. Leaf335 | Transposon recipient (Actinobacteria) | NB | 30 °C | no | [19] |
| *Arthrobacter* sp. Leaf145 | Transposon recipient (Actinobacteria) | NB | 30 °C | no | [19] |
| *Microbacterium* sp. Leaf320 | Transposon recipient (Actinobacteria) | R2A | 30 °C | no | [19] |
| *Microbacterium* sp. Leaf347 | Transposon recipient (Actinobacteria) | R2A | 30 °C | no | [19] |
| *Plantibacter* sp. Leaf1 | Transposon recipient (Actinobacteria) | NB | 30 °C | no | [19] |
| *Rathayibacter* sp. Leaf296 | Transposon recipient (Actinobacteria) | NA | 30 °C | no | [19] |
| *Rhodococcus* sp. Leaf225 | Transposon recipient (Actinobacteria) | NA | 30 °C | no | [19] |
| *Williamsia* sp. Leaf354 | Transposon recipient (Actinobacteria) | NB | 30 °C | no | [19] |
| *Acidovorax* sp. Leaf84 | Transposon recipient (Proteobacteria) | R2A | 30 °C | yes | [19] |
| *Sphingomonas melonis* FR1 | Transposon recipient (Proteobacteria) | NB | 30 °C | yes | [20] |
| *Pantoea eucalypti* 299R | Transposon recipient (Proteobacteria) | LB | 30 °C | yes | [21] |
| *Pseudomonas syringae* B728a | Transposon recipient (Proteobacteria) | LB | 30 °C | no | [22] |
| *Sphingomonas* sp. Leaf17 | Transposon recipient (Proteobacteria) | NB | 30 °C | no | [19] |
| *Sphingomonas* sp. Leaf34 | Transposon recipient (Proteobacteria) | R2A | 30 °C | no | [19] |
| *Sphingomonas* sp. Leaf357 | Transposon recipient (Proteobacteria) | R2A | 30 °C | yes | [19] |
| *Pedobacter* sp. Leaf194 | Transposon recipient (Bacteroidetes) | R2A | 30 °C | no | [19] |

n.a.: Not applicable.

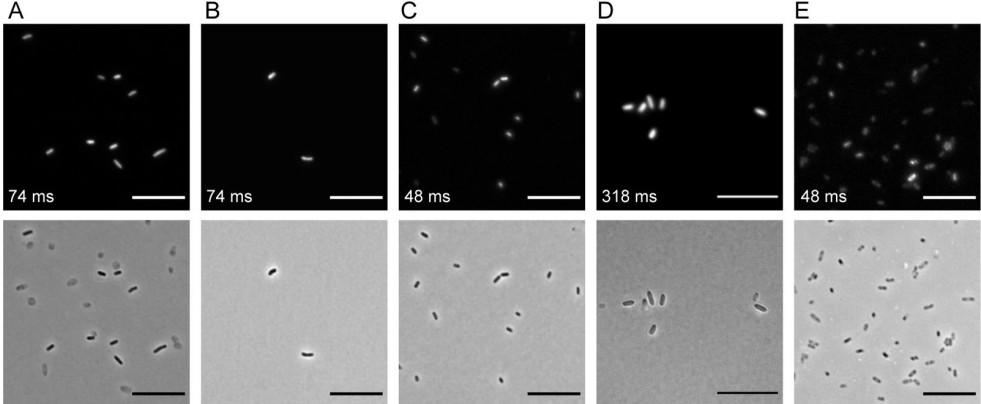

**Figure 2.** Fluorescence of recipients after stable integration of constitutively expressed fluorescent protein genes at the single cell resolution. Representative images of fluorescently tagged bacterial strains. In all cases, fluorescence (upper panel) and phase contrast images (lower panel) are included. (**A**) *Acidovorax* sp. Leaf84::MRE-Himar-145. (**B**) *Sphingomonas* sp. Leaf357::MRE-Himar-145. (**C**) *Sphingomonas melonis* FR1::MRE-Himar-156; (**D**) *Pantoea eucalypti* 299R::MRE-Himar-145; (**E**) *Escherichia coli* ST18 (pMRE-Himar-156). Exposure times are given in the respective fluorescence images. Scale bar = 10 μm.

**Table 3.** Transposon insertion sites.

| Strain Name | Transposon Insertion Flanking Region Sequence | Region Hit |
|---|---|---|
| *Sphingomonas* sp. Leaf357::MRE-Himar-145/1 | AGGGGCTCGCAGTCGATTTACCGGTTCGCATGATCGTAACCGCACAGGG GAAGGAAACATGGGCTCTCTTCCGCCAGCGCGGTGGGATGTACCCTGAG | Beta-hexosaminidase CDS |
| *Sphingomonas* sp. Leaf357::MRE-Himar-145/2 | TGTTATAACCCGGGGCCCAGAAGCGCGAGGTAGTCTTTGAATGGATA CATGGGCAGATATGCGATAAGCCGTCGAGCTTCCGGTTGGCGACGTCT CAGTCCGCGTCCATGACGACCCCGAGCGTGT | Hypothetical protein |
| *Sphingomonas* sp. Leaf357::MRE-Himar-145/4 | CTCGGCGCGCAGGCCAATCTGTGGGCCGAATATATCGTGACGCCCACCGAA TCCCAACATGCGCTGTTCCCGCGCGTCGACGCGCTGGCCGAGATCGCCTG | Hypothetical protein |
| *Acidovorax* sp. Leaf84::MRE-Himar-145/2 | AGTCAACATCGAAAAGCTCGGAGACTATGTGAATCGCTATGGCGTCAATA GCTTTTTCGACGCATCCGATGATGCCCATC | Intergenic region |
| *Acidovorax* sp. Leaf84::MRE-Himar-145/5 | ATCTGATCTTCAGACAGTCTGTCGGTAGCTCCCTCGCGCCTTGCAGAGC AGATGATGTGTTCCCCTTGAAAAACGCCCTTGACATCATGCACCTCGACG | Hypothetical protein |
| *Acidovorax* sp. Leaf84::MRE-Himar-145/6 | TAATCGGTGGATGGTAAATAGATAGGAAATTTATCACTGTGTTTCATAACA GGTTG | Intergenic region |

### 3. Discussion

A range of new plasmids was constructed for the Chromatic Bacteria toolbox based on the R6K origin of replication plasmid backbone containing the Himar1 transposase. Additionally, a series of plasmids containing the Erm$^R$ gene was constructed. A full list of plasmids constructed is included in Table 1. The previous and novel antibiotic resistance gene combinations make the plasmids a versatile tool for bacterial genetic manipulation that accounts for different antibiotic resistances that naturally occur in the recipient organism. Even though we constructed these additional new plasmids in *E. coli* S17-1 [1,17], we made use of *E. coli* ST18 as a donor strain for conjugation experiments [18]. *E. coli* ST18 is an S17-1 derivative that lacks the *hemA* gene. Mutation in *hemA* results in a strong auxotrophy and dependency of *E. coli* ST18 on exogenously provided 5-aminolevulinic acid. Thereby, there is no need to counterselect against the conjugation donor after conjugation by using minimal media or intrinsic antibiotic resistances of the host. The transposon mutants can be selected on their respective optimal complex medium with the addition of an antibiotic selecting for the transposon. Since the vector contains the R6K origin of replication, it only replicates if the *pir* gene is present and expressed in the host, i.e., the plasmids are suicide vectors that are not able to replicate in their recipient. The successful use of Erm$^R$ as a selective marker has been previously reported in clinically relevant Gram-positive pathogens, such as *Staphylococcus aureus* [23] and *Clostridium perfringens* [14], and lactic acid bacteria, such as *Streptococcus mutans* UA159 [13]. Hence, plasmids containing Erm$^R$ were added as an alternative antibiotic resistance gene, for strains in which the original genes may not be expressed or function correctly.

Using the newly developed vectors, we were able to deliver Himar transposons randomly into bacterial recipients, similarly to the previously described pMRE-Tn5-transposon series [1]. In comparison to our pMRE-Tn5-transposon plasmids, we were able to genetically modify additional recipients, such as *Acidovorax* sp. Leaf84, and *Sphingomonas* sp. Leaf357, while retaining the ability to deliver Himar transposons into *S. melonis* Fr1 and *P. eucalypti* 299R. The enhanced spectrum of successful transposition can likely be accredited to the properties of the Himar1 transposase that does not require additional host cofactors to function [8]. However, we were not successful in conjugating the plasmids and inserting the transposons into the genomes of non-model Gram-positive bacterial strains including *Aeromicrobium* sp. Leaf245, *Agreia* sp. Leaf 335, *Arthrobacter* sp. Leaf145, *Microbacterium* sp. Leaf320, *Microbacterium* sp. Leaf 347, *Plantibacter* sp. Leaf1, *Rathayibacter* sp. Leaf296, *Rhodococcus* sp. Leaf225, or *Williamsia* sp. Leaf354. In the future, we plan to use alternative techniques such as electroporation to deliver Himar transposons into non-model Gram-positive bacteria. However, establishing successful transformation protocols for non-model bacteria requires a high degree of optimisation, and was, therefore, not a part of this study. In the future, it would be interesting to benchmark this activity against other transposition systems, such as the Phage Mu transposition complex which has successfully been used to generate insertion libraries in Gram-negative and Gram-positive bacteria [23,24]. During such tests, it may be beneficial to further explore the Tn5 transposome system [25], which works using a similarly to the Phage Mu transposition complex by using transposase proteins that are pre-attached to a linear insertion sequence. Although the Tn5 mediated insertion using the expression of transposon genes previously tested by Schlechter et al. (2018) [1] was unsuccessful in tagging Gram-positive bacteria, the use of a transposome system may be more efficient [26,27].

To map the insertion site of the transposons, arbitrary PCR can be used as described above [1]. Alternatively, the R6K origin of replication is present in the transposon which allows, next to the above-described arbitrary PCR, determination of the insertion site of the transposon into the recipient's genome. To that end, the genome of the recipient can be isolated, digested with a rare restriction enzyme that does not cut in the transposon sequence, such as KpnI, and then re-ligated and transformed into *E. coli* S17-1 or another λpir factor expressing *E. coli* cloning host [28]. Due to the R6K origin of replication contained and the antibiotic resistances located in the transposon, this will result in functional plasmids.

These plasmids can then be sequenced using a sequencing primer (Table 4) to determine the sequence of the insertions site. This process is significantly more time intensive and less cost effective than the arbitrary PCR described above, but might be advantageous in cases where the arbitrary PCR does not yield results of sufficient quality.

**Table 4.** Primers used in this study.

| Name | Sequence (5′ to 3′) [1] | $T_m$ (°C) |
|---|---|---|
| pHimarEm1+XmaI_FWD | cccgggCAATTCGAGGGGTATCGCTCT | 67 |
| pHimarEm1_XbaI_RVS2 | tctagaGCACGAGGAAATTGCGCAAAAA | 67 |
| pMRE-HimarEm1+XbaI_overhang_FW2 | gcaatttcctcgtgctctagaATATAAACTGCCAGGAATTGGG | 62 |
| pntpII_1_REV | GCCATGTAAGCCCACTGCAAGCTAC | 73 |
| pntpII_1_FWD | GTAGCTTGCAGTGGGCTTACATGGC | 73 |
| pMRE-HimarEm1+XmaI_overhang_RV | gatacccctcgaattgcccgggCTGGCGGCCGCAAGCTCC | 74 |
| pMRE-HimarEm1_EmR+XbaI_overhang_RV | tttcatccttcgtagtctagaCATAAACTGCCAGGAATTGGGGAT | 67 |
| pHimarEm1_EmR+XbaI_RV | tctagaCTACGAAGGATGAAATTTTTCAGGG | 63 |
| ARB-RB-PCR1 | CTGGGGTAATGACTCTCTAGC | 59 |
| ARB-RB-PCR2 | CTGAGTAGGACAAATCCGCCG | 62 |
| PCR2 AP-PCR | GGCCACGCGTCGACTAGTCA | 66 |
| Arb1 | GGCCACGCGTCGACTAGTCANNNNNNNNNNGCTCG | n.a. |
| Arb2 | GGCCACGCGTCGACTAGTCANNNNNNNNNNGACTC | n.a. |
| Arb3 | GGCCACGCGTCGACTAGTCANNNNNNNNNNGATAC | n.a. |
| Sequencing primer | CTGGTTCCGCGCACATTTC | 61 |

[1] Primer sequences in majuscules indicate base complementarity to the PCR template, and minuscules indicate overhanging regions with complementarity to adjacent fragments. n.a.: not applicable.

As described in Schlechter et al. (2018), the here-constructed plasmids allow for convenient fluorescent tagging of environmental bacteria and cover a different host range compared to the previously described vectors. Fluorescent protein tags are the prerequisite for many experimental studies that follow different populations simultaneously, identify focal populations in complex environments, or to follow the behaviour of individual cells [4,29–32]. Currently, many delivery systems, including the first versions of the Chromatic bacteria, function almost exclusively in Proteobacteria; the Himar transposons have been shown to have a wider range of activity. Thereby the here-introduced plasmids can serve as a one fits many solutions for tagging Proteobacteria.

## 4. Materials and Methods

### 4.1. Strains and Media

All strains, respective growth media, and growth temperature used in this study are listed in Table 2. Lysogeny broth (LB, HiMedia, Kuwait City, Kuwait), nutrient broth (NB; HiMedia), Reasoner's 2A media (R2A, HiMedia), and Reasoner's 2A agar (R2A agar, HiMedia) were prepared according to manufacturer's instructions. Media were supplemented with 1.5% Oxoid™ bacteriological agar (Agar No. 1, Thermo Fisher, Waltham, MA, USA) where needed. To support growth of *E. coli* ST18, media was supplemented with 50 mg L$^{-1}$ 5-aminolevulinic acid (5-ala, Sigma, St. Louis, MO, USA). When appropriate, media were supplemented with antibiotics with the following concentrations: 100 mg L$^{-1}$ ampicillin, 20 mg L$^{-1}$ chloramphenicol, 10 mg L$^{-1}$ colistin, 100 mg L$^{-1}$ erythromycin, 15 mg L$^{-1}$ gentamicin, and/or 50 mg L$^{-1}$ kanamycin.

### 4.2. Plasmid Construction

The Himar transposase, antibiotic resistance, and fluorescent protein genes were retrieved from the plasmids pHimarEm1, pMRE13X, pMRE14X, and pMRE15X to construct the Himar-based transposon delivery plasmid series [1,12]. Plasmids were constructed as previously described [1]. The here-described plasmids are denoted pMRE-Himar-1XY series, where X can either be 3, 4, 5, 7, or 9, representing the different antibiotic resistance gene combinations of chloramphenicol; gentamicin and chloramphenicol; kanamycin and chloramphenicol; erythromycin and chloramphenicol, or erythromycin, kanamycin, and

chloramphenicol, respectively, and where Y can be 0, 1, 2, 3, 4, 5, 6, or 7 representing blue, cyan, green, yellow, orange, red, near-infrared, or green fluorescent protein (FP) gene, respectively (mTagBFP2, mTurquoise2, sGFP2, mOrange2, sYFP2, mScarlet-I, mCardinal, or mClover3, respectively).

Each plasmid was constructed by amplifying three fragments: the plasmid backbone, a unique antibiotic resistance gene (X fragment), and a unique FP gene upstream of a chloramphenicol resistance gene (Y fragment). All PCRs were performed using Phusion High-Fidelity Polymerase (Thermo Scientific). Primers used for cloning are listed in Table 4. For PCR mixes containing primers with overhangs, a touchdown PCR protocol was performed starting with an annealing temperature 10 °C above the lowest $t_m$ of the primer pair. This was reduced by 1 °C for ten cycles before running the PCR with the annealing temperature set to $t_m$. After amplification, all PCR reactions were DpnI treated to digest the methylated plasmid template DNA before the PCR fragments were purified with the DNA Clean & Concentrator Kit (Zymo Research, Irvine, California). To construct pMRE-Himar-13X, 14X and 15X, the plasmid backbone was amplified from pHimarEm1 using pHimarEm1+XmaI_FWD and pHimarEm1_XbaI_RVS2. Antibiotic resistance genes and fluorescent protein genes were amplified from pMRE-13X, pMRE-14X, or pMRE15X. The X fragments were amplified using primers pMRE-HimarEm1+XbaI_overhang_FW2 and pntpII_1_REV. The Y fragments were amplified using primers pntpII_1_FWD and pMRE-HimarEm1+XmaI_overhang_RV. For construction of the pMRE-Himar-17X and pMRE-Himar-19X series, pHimarEm1 was amplified using primers pHimarEm1+Xmal_FWD and pHimarEm1_EmR+XbaI_RV. Using primers pMRE-HimarEm1_EmR+XbaI_overh_FWD and pntpII_1_REV, antibiotic resistance genes were amplified from pMRE-13X to create the pMRE-Himar-17X series, or from pMRE-15X to create pMRE-Himar-19X series. The FP fragment was amplified using the primers pntpII_1_FWD and pMRE-HimarEm1_EmR+XbaI_overh_RV from pMRE-15X plasmids.

Gibson assembly was performed as previously described [1]. Briefly, the fragments were mixed at a 1:3 backbone: insert molar ratio with between 20–100 ng of backbone fragment being used. No more than 5 µL DNA solution was added to a 15 µL Gibson assembly mix. Where appropriate, water was added to top up the reaction volume to 20 µL. The Gibson assembly mix was incubated at 50 °C for 20 min. Chemically competent *E. coli* S17-1 cells were then transformed using 10 µL of the Gibson assembly mix [33]. After the plasmids were confirmed by Sanger sequencing, they were cloned into chemically competent *E. coli* ST18 [33].

*4.3. Transposon Delivery Using Conjugation*

Two parental matings were performed to deliver the pHimar1Em-based plasmids into a range of bacterial strains, following the protocol described by Schlechter et al. (2019) [34]. In variation to this protocol, conjugations were performed using the auxotrophic *E. coli* ST18 as a plasmid donor strain [18]. To perform the conjugations, recipient strains (Table 2) were grown in 50 mL of suitable media for up to three days depending on their growth rate. Single colonies of *E. coli* ST18 donor strains were produced and used to inoculate overnight cultures of LB supplemented with appropriate antibiotics and 5-aminolevulinic acid (5-ala). From these overnight cultures, 2 mL was inoculated into 100 mL fresh LB and grown until the donor cultures reached an $OD_{600nm}$ of 0.5. Then, both donor and recipient cells were collected by centrifugation and resuspended in phosphate buffered saline (PBS) to an $OD_{600nm}$ of 1. Approximately 5 mL recipient and donor cells were combined in a 1:1 ratio, then centrifuged and resuspended in 200 µL of PBS. The suspension was spotted onto a 0.44 µm S-pak Membrane filter (Millipore) which was placed on LB agar plates supplemented with 5-ala. The bacterial mixes were incubated at 30 °C overnight. Bacteria were recovered from the filter by vigorous vortexing in 10 mL PBS in 50 mL falcon tubes. Subsequently, the filter was dismissed, and the mixes were concentrated by centrifugation and resuspension in 1 mL PBS, before 10 µL, 100 µL, and the remaining volume were plated on media without 5-ala and appropriate antibiotics to select for transconjugants.

Transposon insertion mutant colonies appeared up to five days after conjugations were performed. To obtain a pure culture of the transposon insertion mutants, colonies were restreaked at least three times.

*4.4. Screening for Fluorescent Colonies and Fluorescence Microscopy*

For convenient and quick assessment of colony level fluorescence, a blue light gel reader was used for fluorescent proteins with emission wavelengths between 500 and 680 nm. Fluorescence microscopy was performed on a Zeiss AxioImager.M1 fluorescent widefield microscope equipped with Zeiss filter sets 38HE, 43HE, 46HE, and 47HE, (BP 470/40-FT 495-BP 525/50, BP 550/25-FT 570-BP 605/70, BP 500/25-FT 515-BP 535/30, and BP 436/25-FT 455-BP 480/40, respectively), an Axiocam 506, and the software Zeiss Zen 2.3. Single-cell fluorescence was analysed as described previously [35]. In short, bacteria were mounted on an agarose slab (~1 mm thick, 1% agarose in milliQ water) and samples were analysed using a Zeiss AxioImager.M1 at 1000× magnification. Images were processed using ImageJ/Fiji [36].

*4.5. Arbitrary PCR*

The transposon insertion of fluorescently (FP)-tagged bacterial strains was mapped using arbitrary PCR [37]. Briefly, a first PCR was performed using a mix of random primers Arb1, Arb2, and Arb3 containing an adapter oligo at the 5'-end and a primer targeting the transposon insertion (Table 4). Amplification was performed in a total volume of 25 μL containing: 15.05 μL ddH$_2$O, 5 μL Phusion GC buffer, 1.25 μL 10 mM dNTP mix, 1.25 μL 10 μM arbitrary primers, 0.5 μL 10 μM ARB-RB-PCR1, 0.75 μL DMSO, 0.2 μL Phusion, and 1 μL DNA template. Cycling conditions were as follows: initial denaturation of 95 °C for 5 min; six cycles of 30 s at 95 °C for denaturation, 30 s at 30 °C for annealing, and 1.5 min at 72 °C for extension; then, 30 cycles of 30 s at 95 °C for denaturation, 30 s at 45 °C for annealing, 2 min at 72 °C for extension, and 5 min at 72 °C for a final extension. Then, PCR products were purified using DNA Clean & Concentrator$^{TM}$-5 (Zymo Research) following the manufacturer's recommendations and eluted in 30 μL. A second PCR was performed using primers PCR2-AP and ARB-RB-PCR2. Amplification was performed in a total volume of 25 μL, containing: 12.8 μL ddH$_2$O, 5 μL Phusion GC buffer, 1.25 μL 10 mM dNTP mix, 1 μL 10 μM arbitrary primers, 1 μL 10 μM ARB-RB-PCR1, 0.75 μL DMSO, 0.2 μL Phusion, and 3 μL of ten-fold dilution of purified PCR fragments. Cycling conditions were as follows: initial denaturation of 95 °C for 3 min; 30 cycles of 30 s at 95 °C for denaturation, 30 s at 52 °C for annealing, 2 min at 72 °C for extension, and 5 min at 72 °C for a final extension. PCR products were then purified and concentrated using the Zymo Research DNA Clean & Concentrator$^{TM}$-5 following the manufacturer's recommendations and sequenced using Sanger's sequencing (Macrogen, Seoul, Korea). The sequencing results were mapped against the genome of the corresponding bacterial strain using Blast [38] and a local database of draft genome sequences of the strains used in this study using the software Geneious Prime (version 2020.1.2, Biomatters, Auckland, New Zealand).

**5. Conclusions**

The here-described plasmids extend the previously constructed Chromatic bacteria toolbox host range and utility. All plasmids and sequences will be made available through the non-profit service addgene.org upon peer reviewed publication (Addgene plasmids numbers to be determined). The extended Chromatic bacteria vector series can often be used as a one-stop solution for fluorescently marking bacteria to enable a hassle-free solution for groups that do not want to try and establish novel genetic systems for newly isolated bacterial strains.

**Author Contributions:** Conceptualization, R.O.S. and M.N.P.R.-E.; methodology, C.S., R.O.S., and M.N.P.R.-E.; formal analysis, C.S.; investigation, C.S.; writing—original draft preparation, C.S.; writing—review and editing, C.S., R.O.S., and M.N.P.R.-E.; visualization, C.S. and R.O.S.; supervision, R.O.S. and M.N.P.R.-E.; project administration, M.N.P.R.-E.; funding acquisition, M.N.P.R.-E. All authors have read and agreed to the published version of the manuscript.

**Funding:** This project was supported by the New Zealand Tertiary Education Commission CoRE grant to the Bio-Protection Research Centre and a Marsden Fast-Start grant to M.N.P.R-E (UOC1704) and additional funding by Callaghan Innovation is acknowledged. C.S. is supported by a UC Masters scholarship and a Bio-Protection Research Centre summer scholarship. R.O.S. is supported by a New Zealand International Doctoral Research Scholarship (NZIDRS) and a University of Canterbury Doctoral Scholarship.

**Institutional Review Board Statement:** Not applicable.

**Informed Consent Statement:** Not applicable.

**Data Availability Statement:** Plasmids will be made available at https://www.addgene.org/Mitja_Remus-Emsermann/, accessed on 7 December 2021.

**Acknowledgments:** The authors thank Mark Silby and Lucy McCully (University of Massachusetts, Dartmouth) for the kind gift of pHimarEm1. The publication of this article was funded by Freie Universität Berlin.

**Conflicts of Interest:** The authors declare no conflict of interest. The funders had no role in the design of the study; in the collection, analyses, or interpretation of data; in the writing of the manuscript; or in the decision to publish the results.

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
