# Peer review of "Chromatic Bacteria v.2-A Himar1 Transposon-Based Delivery Vector to Extend the Host Range of a Toolbox to Fluorescently Tag Bacteria"

_2674-1334, doi:10.3390/bacteria1010006_

Round 1
Reviewer 1 Report
The manuscript entitled” Chromatic bacteria v.2 - A Himar1 transposon-based delivery vector to extend the host range of a toolbox to fluorescently tag bacteria” developed a new toolkit, which was improved based on the authors’ previous work, by harnessing Himar1 transposon to fluorescently tag a variety of bacterium species. Compared with its former version, this technological advancement enables a broader range gene manipulation within environmental bacterium species, not restricted to proteobacteria, but also independent from introducing host-specific factors other than the presence of TA dinucleotide sequences across the genome. In my opinion, the manuscript is suitable for publication, after the authors have addressed the following comments and questions.
Major points:
1 I am not sure if I fully understand the necessity of introducing erythromycin as additional selectable markers. Can the authors elaborate on this?
2 Despite the wide ranges of bacterial species that could be genetically modified via this technique, have the authors compared the differences amongst said species in terms of base composition, eg, GC content%, or gene size? Have the authors checked any bias due to these differences that may lead to the uneven efficiencies of integration? If so, how big the influence will be? One way to find out is to use Tnseq estimated based on the read counts or template counts corresponding to an insertion site.
3 In the previous paper the authors extensively tested the emission spectra and intensity of all fluorescent proteins introduced here. Have they tested at least these parameters in this system too? Will some combinations result in significantly brighter signals that would be preferable to apply than the others?
4 Figure 2: Due to the lack of controls, it would be hard to discern whether all bacilli displayed within this selected field are fluorescently tagged. For example, in panel A, C and E some single cells generate dimmer signals. Are those self-luminescent or due to differential expression of targeted genes?
Minor points:
1 Figure 1 illustrations: It is very confusing to label plasmids in panel A and B with the same nomenclature. 13Y/15Y vs. 17Y/19Y. Pls correct the labels. Besides, could they highlight Erm resistant gene?
2 Figure 2: In its legend and photos, the authors should label or depict the exposure time during image acquisition. Otherwise, it would be very confusing.
Author Response
The manuscript entitled” Chromatic bacteria v.2 - A Himar1 transposon-based delivery vector to extend the host range of a toolbox to fluorescently tag bacteria” developed a new toolkit, which was improved based on the authors’ previous work, by harnessing Himar1 transposon to fluorescently tag a variety of bacterium species. Compared with its former version, this technological advancement enables a broader range gene manipulation within environmental bacterium species, not restricted to proteobacteria, but also independent from introducing host-specific factors other than the presence of TA dinucleotide sequences across the genome. In my opinion, the manuscript is suitable for publication, after the authors have addressed the following comments and questions.
Major points:
1 I am not sure if I fully understand the necessity of introducing erythromycin as additional selectable markers. Can the authors elaborate on this?
We gladly elaborate on the addition of erythromycin as a selectable marker. Previously we have provided transposons that deliver chloramphenicol, gentamicin, kanamycin and tetracycline resistance genes. While these resistances can successfully be selected in many Proteobacteria, other bacteria such as Lactic acid bacteria, Pedobacter strains and many others, are often not selectable with these antibiotics. Erythromycin is therefore an alternative to the previously introduced antibiotics. To make this more clear in our manuscript, we have added the following text in Lines 129-134: "The successful use of ErmR as a selective marker has been previously reported in clinically relevant Gram-positive pathogens such as Staphylococcus aureus [29], Clostridium perfringens [14] and lactic acid bacteria such as Streptococcus mutans UA159 [13]. Hence, plasmids containing ErmR were added as an alternative antibiotic resistance gene, for strains in which the original genes may not be expressed or function correctly".
2 Despite the wide ranges of bacterial species that could be genetically modified via this technique, have the authors compared the differences amongst said species in terms of base composition, eg, GC content%, or gene size? Have the authors checked any bias due to these differences that may lead to the uneven efficiencies of integration? If so, how big the influence will be? One way to find out is to use Tnseq estimated based on the read counts or template counts corresponding to an insertion site.
We agree with the reviewer that an extensive study that determines the host range of the Himar transposon in a wide range of bacteria is lacking. While we appreciate the importance of such a study, this is clearly beyond the scope of the presented study. Beyond genome composition, other factors such as the presence of restriction endonuclease, and conjugation competence are factors that may impact the frequency of successful insertion very much.
3 In the previous paper the authors extensively tested the emission spectra and intensity of all fluorescent proteins introduced here. Have they tested at least these parameters in this system too? Will some combinations result in significantly brighter signals that would be preferable to apply than the others?
We were not planning to further characterize the brightness of the different constructs as we did not exchange the promoters driving the expression of the fluorescent protein genes.
4 Figure 2: Due to the lack of controls, it would be hard to discern whether all bacilli displayed within this selected field are fluorescently tagged. For example, in panel A, C and E some single cells generate dimmer signals. Are those self-luminescent or due to differential expression of targeted genes?
Exposure times were included in Figure 2 for each representative picture. Dimmer fluorescence in C and E compared to others could be due to the brightness of the fluorescent protein itself. mCardinal was expected to give a dimmer signal (C and E, mCardinal Extinction coefficient: 87,000 M-1 cm-1; quantum yield: 0.19) compared to mScarlet (Extinction coefficient: 104,000 M-1 cm-1; quantum yield: 0.54).
Minor points:
1 Figure 1 illustrations: It is very confusing to label plasmids in panel A and B with the same nomenclature. 13Y/15Y vs. 17Y/19Y. Pls correct the labels. Besides, could they highlight Erm resistant gene?
As suggested, we have added more specific labels into the plasmids. Additionally, we have added a colour label to the Erythromycin resistance.
2 Figure 2: In its legend and photos, the authors should label or depict the exposure time during image acquisition. Otherwise, it would be very confusing.
The exposure times were added to the images.
Reviewer 2 Report
This is well written paper, and it extends the transposon tool arsenal to many new organisms. I found the text very well crafted with all the details needed for this type of methodology-oriented paper. Although this paper is a sequel to their previous paper, overall describing the system initially, I feel that the new additional tools/results need to be published.
One conceptual addition should be added. The authors cite in their previous paper the Tn5 transpososome-mediated gene delivery, which is efficient with many types of organisms. This should be discussed here in this new paper as well.
In addition, the Mu-transpososome-mediated gene delivery should be discussed and cited, either in the "Intro" or in the "Discussion"; preferably in both of them. This methodology is highly efficient not only for Gram-negatives (1) but also for Gram-positives (2). Mu system does not require any host factors, and it has been used for the generation of exhaustive mutant libraries and functional analyses of genomes both with Gram negatives (3) and Gram positives (4). Although there exists a number of other publications with the system as well, I believe the four papers below portray a minimum set for a comprehensive description.
1. Lamberg, A., et al. 2002: -Appl. Environ. Microbiol. 68:705-712.
2. Pajunen, M. I., et al. 2005: -Microbiology 151: 1209-1218.
3. Laasik, E., et al. 2005: -FEMS Microbiol. Lett. 243: 93-99.
4. Tu Quoc P. H., et al. 2007: -Infect. Immun. 75: 1079-1088.
Author Response
Reviewer 2: Comments and Suggestions for Authors
This is well written paper, and it extends the transposon tool arsenal to many new organisms. I found the text very well crafted with all the details needed for this type of methodology-oriented paper. Although this paper is a sequel to their previous paper, overall describing the system initially, I feel that the new additional tools/results need to be published.
We thank the reviewer for their general support of our manuscript.
One conceptual addition should be added. The authors cite in their previous paper the Tn5 transpososome-mediated gene delivery, which is efficient with many types of organisms. This should be discussed here in this new paper as well.
To accommodate the reviewers request, we have added the following text in lines 155-158: “Although the Tn5mediated insertion using the expression of transposon genes previously tested by Schlechter et al. (2018) [1] was unsuccessful in tagging Gram-positive bacteria, the use of a transposome system may be more successful [32,33].”
In addition, the Mu-transpososome-mediated gene delivery should be discussed and cited, either in the "Intro" or in the "Discussion"; preferably in both of them. This methodology is highly efficient not only for Gram-negatives (1) but also for Gram-positives (2). Mu system does not require any host factors, and it has been used for the generation of exhaustive mutant libraries and functional analyses of genomes both with Gram negatives (3) and Gram positives (4). Although there exists a number of other publications with the system as well, I believe the four papers below portray a minimum set for a comprehensive description.
1. Lamberg, A., et al. 2002: -Appl. Environ. Microbiol. 68:705-712.
2. Pajunen, M. I., et al. 2005: -Microbiology 151: 1209-1218.
3. Laasik, E., et al. 2005: -FEMS Microbiol. Lett. 243: 93-99.
4. Tu Quoc P. H., et al. 2007: -Infect. Immun. 75: 1079-1088.
The reviewer raises an interesting point and we have included the Mu transposon system to the introduction and discussion (Lines 54-57 and 149-155).